# Yeast Cell as a Bio-Model for Measuring the Toxicity of Fish-Killing Flagellates

**DOI:** 10.3390/toxins13110821

**Published:** 2021-11-21

**Authors:** Malihe Mehdizadeh Allaf, Charles G. Trick

**Affiliations:** 1Department of Biology, Western University, London, ON N6A 5B7, Canada; mmehdiz@uwo.ca; 2Interfaculty Program in Public Health, Schulich School of Medicine and Dentistry, Western University, London, ON N6G 2M1, Canada

**Keywords:** bioassay, yeast, harmful algal blooms, *Heterosigma akashiwo*, *Prymnesium parvum*

## Abstract

Harmful algal blooms are a significant environmental problem. Cells that bloom are often associated with intercellular or dissolved toxins that are a grave concern to humans. However, cells may also excrete compounds that are beneficial to their competition, allowing the cells to establish or maintain cells in bloom conditions. Here, we develop a yeast cell assay to assess whether the bloom-forming species can change the toxicity of the water environment. The current methods of assessing toxicity involve whole organisms. Here, yeast cells are used as a bioassay model to evaluate eukaryotic cell toxicity. Yeast is a commonly used, easy to maintain bioassay species that is free from ethical concerns, yet is sensitive to a wide array of metabolic and membrane-modulating agents. Compared to methods in which the whole organism is used, this method offers rapid and convenient cytotoxicity measurements using a lower volume of samples. The flow cytometer was employed in this toxicology assessment to measure the number of dead cells using alive/dead stain analysis. The results show that yeast cells were metabolically damaged after 1 h of exposure to our model toxin-producing euryhaline flagellates (*Heterosigma akashiwo* and *Prymnesium parvum*) cells or extracts. This amount was increased by extending the incubation time.

## 1. Introduction

Harmful algal blooms (HABs) are a major universal threat due to their significant impact on ecosystems, public health, tourism, and fisheries [1]. The frequency of HABs has increased in recent decades as a result of climate change, and cultural eutrophication caused by domestic, industrial, and agricultural waste [2,3,4]. Among the different phytoplankton species responsible for the formation of HABs, flagellate species represent 90% and dinoflagellates represent 75% (45–60 taxa) [5]. 

The harmful species’ mechanisms, which can cause mortality or physiological impairment, can be divided into two general types: mechanical and physical damage, and chemical damage [6]. A non-toxic bloom-forming algal species accumulates a high biomass and results in surface water discoloration, which kills both fish and invertebrates due to oxygen depletion or starvation [2,3,6]. The oxygen depletion may be caused by the high respiration rate of algae during night or in dim light during the day, or by bacteria during the decay of the bloom [2]. Chemical damage happens through the production of potent toxins, which can accumulate or transfer through the food chain to humans via shellfish and fish consumption [2,7,8,9]. The toxin can cause gastrointestinal and neurological illnesses, such as paralytic shellfish poisoning (PSP) [8,9], diarrhetic shellfish poisoning (DSP) [2,4], amnesic shellfish poisoning (ASP) [2,3], ciguatera fish food poisoning [10] and neurotoxic shellfish poisoning (NSP) [9]. 

There are also a limited number of species with HAB attributes that are particularly damaging to fish and invertebrates and, therefore, can cause great economic losses. Genera *Prymnesium*, *Chrysochromulina*, and *Phaeocystis* from prymnesiophytes and *Chattonella*, *Heterosigma*, and *Fibrocapsa* from raphidophytes belong to this category [11].

*Heterosigma akashiwo* and *Prymnesium parvum* are two golden-brown flagellate species from the raphidophyte and prymnesiophyte classes, respectively. These two active fish-killers have the high potential to kill thousands or even millions of fish per bloom. The blooms of *H. akashiwo* are sporadic and have been reported in Canada and the United States [12,13,14,15], Mexico [16], Japan [17,18], Chile [19], China [20], New Zealand [21], and Norway [22]. However, the blooms of *P. parvum,* widely distributed throughout temperate regions, have frequently been reported in Israel [23] England [24], Norway [25], China [26], Tunis [27], and United States [28]. Although these two species are taxonomically separated from each other, they apparently kill fish in a similar fashion.

The toxicity mechanisms of these types of flagellates are controversial and unresolved. Scientists have suggested the following mechanisms are active procedures that kill fish: (a) mucus secretion causes fish asphyxiation by covering the gill cells [21], (b) neurotoxins or cardiotoxins result in respiratory and/or cardiac paralysis [17,29], (c) reactive oxygen species (ROS) production alters gill structure and function [3,30,31], and (d) hemolysis compounds cause blood cell lysis [2,32,33].

To measure the ichthyotoxic effect of these flagellates, the most common bioassay is to use different fish strains or species with different ages (larvae, juveniles, or adults), and varying exposure times. Some of the routinely used species are brine shrimp; *Artemia salina* [34,35,36], Japanese sea bream, *Pagrus major* [17,37,38,39], and yellowtail, *Seriola guinqueradiata* [40,41,42]. Using mice as a bioassay models has been debatable since they were first used, due to unreliability and the associated ethical issues [43,44]. Various cell lines obtained from different animals including a mouse [45], a rat [34], fish gill cells [46,47,48,49], or human tissues such as human hepatoma cell line (HepG2) [50], or human colon carcinoma cell line (Caco-2) [51], are recommended and have been used to measure different types of toxins produced by algal bloom species. Using whole organisms or their cell lines as a bioassay model for ichthyotoxicity measurement obligates prolonged exposure or follow-up periods, and therefore increases the errors in evaluation. Furthermore, the high cost of these types of bioassays is a serious concern [35,52]. Therefore, finding a biological cell model to measure the cytotoxins in a reliable, fast, and inexpensive manner would be very beneficial and highly sought-after.

To that purpose this study has proposed the use of yeast cells (*Saccharomyces cerevisiae*), which are widely used as eukaryotic model organisms [53]. The culturing and maintenance of *Saccharomyces cerevisiae* is simple and rapid, and is also free of any ethical objections. In addition, *Saccharomyces cerevisiae* is sensitive to a broad range of toxic substances and economically affordable [54].

## 2. Results

After measuring the percentage of dead yeast cells in the presence of two different fish-killing flagellates; *H. akashiwo* and *P. parvum*; the results revealed that the toxicity effect of these two strains was improved by rupturing the cells using sonication and centrifugation techniques (Figure 1). In *H. akashiwo*, the highest level of toxicity was observed in sample G, which was obtained by sonicating the pellets after centrifugation and an additional sonication step were conducted (Figure 1). The yeast mortality was about 37% after an hour, which increased to 62% after 3 h of incubation. At that stage, the highest level of toxicity in this flagellate belonged to samples I (sonicated pellets after one more centrifugation and sonication step), D (sonicated pellets) and E (supernatant after one sonication and centrifugation step). These fractionated samples killed almost 30% of yeast cells after one hour of exposure. Increasing the incubation time from 1 h to 3 h displayed a significant effect on the mortality of the yeast cells (*p* < 0.001) (Table 1). Moreover, a significant difference in yeast mortality was observed between the control and different treatments (*p* < 0.001). The positive control was prepared by sonicating the yeast cells at a continuous output of nine for a minute and the negative control was prepared by mixing yeast cells with artificial sea water (ASW). Sample G shows a significant difference compared to other treatments, while there was no significant difference between I, D and E.

The highest levels of toxicity in *P. parvum* were detected in the sample treatments of E, D and G, respectively, while there was a significant difference between these sample treatments and the rest of samples (*p* < 0.001). The yeast mortality was more than 35% in this treatment after one hour, which was amplified to almost 50% after 3 h of incubation. Furthermore, the yeast mortality showed a significant effect between positive control and different algal samples (*p* < 0.001). The two-way ANOVA parameters for both species are presented in Table 1.

Among the four highest treatments (D, E, G, I), between *H. akashiwo* and *P. parvum,* no significant difference was observed (Figure 2). However, a significant difference was detected among different treatments (*p* < 0.005). Sample G, prepared from *H. akashiwo*, was significantly different from D, E and I, while sample I, prepared from *P. parvum,* showed the highest difference compared with sample D and E. Samples D and E in *P. parvum* showed more toxicity compared to *H. akashiwo*, while this was vice versa for the other two treatments. There was also a statistically significant difference in interaction between algal species and the four treatments (*p* = 0.001) (Table 2).

To validate the data *Tetraselmis chuii*, a nontoxic marine species, were used in this study (data not shown). Additionally, the effect of salinity and temperature on the mortality of *S. cerevisiae* was measured (unpublished data).

## 3. Discussion

Measuring the level of toxicity in fish-killing flagellates is a complicated process, and not a single standard has been used by the researchers. The floating carcasses of caged fish, or the embayment of fish in the bloom regions, are the commonly accepted signs of the presence of harmful algal blooms. The toxin mechanism framework for fish-killing flagellates such as *H. akashiwo* and *P. parvum* is the production of excessive mucus by microalgae [21], production of brevetoxin-like compounds [17,29], the ability to produce and excrete reactive oxygen species (ROS) [30,31,49], and the production of hemolytic compounds [33].

Alternate approaches that correlate with the production of toxicity have been used by researchers. Some alternate methods for toxicity measurement use a whole organism or larva, hemolytic activity, neurological damage using zebra fish embryos, gill cell degeneration, and cellular permeability. In this study, we propose yeast as a bio- model to measure toxicity in fish-killing flagellates. This method has many benefits over traditional methods. This method is fast and reliable, inexpensive, many samples can be tested in a short period of time, and a small volume of samples is required.

The present study demonstrates that yeast cells can be used as a biological cell model to measure the toxicity of harmful algal species such as *H. akashiwo* and *P. parvum*. The obtained data also revealed that the toxin produced by the aforementioned species was released upon the cell damage of both species. Thus, the suggestion is that the toxic compounds are within certain intracellular compartments, and leakage or release of these compounds from *H. akashiwo* and *P. parvum* cells is enhanced by the breaking of cells. Among different treatments that were proposed in this study, samples D, E, G, and I showed the highest level of yeast mortality.

Similar results were observed for the same species or similar ones using other toxicity measurements. Kuroda et al. [55] stated that the ruptured cell suspension of *Chattonella marina* produced strong hemolytic activity towards rabbit erythrocytes; however, either intact cell suspension or the cell-free culture supernatant did not express any hemolytic activity. The same results were obtained for the extracts of various *Fibrocapsa japonica* strains collected worldwide [56]. Ling and Trick [33] reported that after they fractioned the *H. akashiwo* cells, cell hemolytic activity improved in comparison with the intact cells. Their data indicated that the hemolytic agents were released upon the cell damage of *H. akashiwo* by employing sonication or centrifugation.

Mohamed and Sheheri [36] used aqueous and methanol extracts of *H. akashiwo* to expose to nauplii of 48 h hatched cysts of *Artemia salina* and reported the mortality after 48 h of incubation. Both samples were toxic towards *A. salina*; however, the methanol extracts were more toxic. The same results were also observed for hemolytic activity.

Segar et al. [57] studied the effect of nutrient repletion on the toxicity of *P. parvum*. They observed that lysing *P. parvum* cells through sonication significantly decreased the viability of gill cells and improved the toxicity level of this species in comparison with live cultures.

Our results agree with previous research studies in this field. Our findings suggest that a toxin, or toxins, are kept within cellular compartments of the cells and therefore cellular membrane damage can facilitate and enhance a toxin release from fish-killing species.

Employing this method will allow us to measure the toxicity effect of fish-killing flagellates in a fast and reliable method, while saving time, resources, and money, all of which are crucial to the environment, aquaculture facilities, and public health.

## 4. Conclusions

The fundamental purpose of this research was to find a fast, reliable, and inexpensive method of measuring and evaluating the levels of toxicity within different strains of algal bloom forming species. For this purpose, a yeast cell was recommended for use as a bio-model since they are free of ethical concerns, are sensitive to a range of various toxic elements, are low-cost, and are simple to work with and maintain within a short time span. Performing this assay requires a low volume of samples. In addition, using a flow cytometry technique for measuring the viability of the cells quickens the toxicity measurement process.

The obtained data revealed that breaking the toxic algal cells improved the ichthyotoxic effect of these microorganisms, which means the toxin is located inside the cellular compartments. Sample D, E, G, and I, for both species, showed the highest level of yeast cell mortality. Moreover, the results of the two fish-killing species, with different taxonomic groups, are similar.

*Saccharomyces* includes various species with a similar morphology and the potential to be used as a cell bio-model for measuring toxicity, although further investigation is required.

## 5. Materials and Methods

A non-axenic strain of *Heterosigma akashiwo* (NWFSC-513), isolated in 2010 from Clam Bay, WA, USA, and *Prymnesium parvum* (UTEX LB 2827) isolated near Charleston, SC, USA in 2002 and purchased from the UTEX collection (University of Texas at Austin, Austin, TX, USA), were used in this study.

The stock cultures for *H. akashiwo* and *P. parvum* were maintained in f/2 and f/20 (minus Si) medium [58], respectively, complemented by artificial seawater medium (ESAW) [59]. The cultures were grown in a 250 mL Erlenmeyer flask at 20 ± 1 °C and under a continuous white fluorescent light intensity of 80 ± 5 µmol photons m^−2^ s^−1^.

*Saccharomyces cerevisiae* was grown for 15 ± 1 h at room temperature in YPD medium (1% yeast extract, 2% peptone, 2% dextrose, 2% agar (all *w*/*v*)).

To measure the toxicity of both strains, algal samples were prepared from exponentially growing cultures (4–5 days after inoculation). All samples and experiments were performed in triplicate. The data were presented as an average value ± standard deviations.

The algal treatment samples were prepared as follows:Sample A: Intact cells (viable cells and extracellular material) of both species.Sample B: Ultrasonic rupture cells suspensions, which were obtained by sonicating 7 mL culture suspensions in an ice bath with a continuous output power of 9 for 1 min with a Virsonic 100 Ultrasonic Cell Disrupter (VirTis Company, Gardiner, NY, USA).Sample C: culture supernatants were prepared by centrifuging a 7 mL sample A at 500× *g* for 10 min at 4 °C, using 15 mL falcon centrifuge tubes in a Beckman GH-3.8/GH-3.8A swing-out rotor (Beckman Coulter, Fullerton, CA, USA).Sample D: resuspended pellets in ASW and sonicated for 1 min with a continuous output power of 9.Sample E: Sample D supernatants, were prepared by centrifuging sample D at 6100× *g* for 15 min at 4 °C.Sample F: Sample B supernatants, were obtained by centrifuging sample B at 500× *g* for 10 min at 4 °C.Sample G: resuspended pellets from sample B and sonicated with similar output power for sample B and D.Sample H: Sample F supernatants, were attained by centrifuging at 6100× *g* for 15 min at 4 °C.Sample I: resuspended, sonicated pellets from sample F, with similar power for pervious samples.Negative control was prepared by mixing yeast with ASW.Positive control was prepared by sonicating the yeast sample for 1 min with a continuous output power of 9.

The algal sample preparation procedure is outlined in Figure 3. To measure toxicity, the prepared algal samples were combined with the yeast cells with a ratio of 5 to 1 and incubated for 1 and 3 h at room temperature. To measure the number of cells and fluorescence intensity, a Turner Designs PhytoCyt flow cytometer (Sunnyvale, CA, USA) was used. To measure the background, the samples were run without dye. Then, 15 min before toxicity measurement, a final concentration of 1.5 µM SYTOX^®^ Green (Life Technologies, Carlsbad, CA, USA), a high-affinity nucleic acid stain that permeates into the cells with compromised plasma membranes [60], was added to measure the cell integrity and the percentage of dead cells. The fluorochrome SYTOX^®^ Green was employed to estimate the degree of cell membrane permeability of yeast during treatments. The highest toxicity should be expressed in samples when the cells are most permeable, as indicated by the greatest level of SYTOX^®^ Green fluorescence per cell.

To determine the live/dead cells using SYTOX^®^ Green fluorescence dye, CFlow^®^ Plus software, version 1.0.227.5 was used. A bivariate scatter plot with green versus chlorophyll a fluorescence detectors was applied. The plot was divided into four quadrants where the unstained cell population was placed in the lower left quadrant, while the stained cell population, which represented the dead or compromised cells, was in the upper right quadrant.

Statistical analysis: Data reporting the percentage of dead yeast cells are expressed as mean ± SD (n = 3). The data were compared using a two-way analysis of variance (ANOVA) followed by Tukey multiple comparison tests analyzed by SigmaPlot 12.0. A significance level of 95% (α = 0.05) was considered for all statistical analyses.

## Figures and Tables

**Figure 1 toxins-13-00821-f001:**
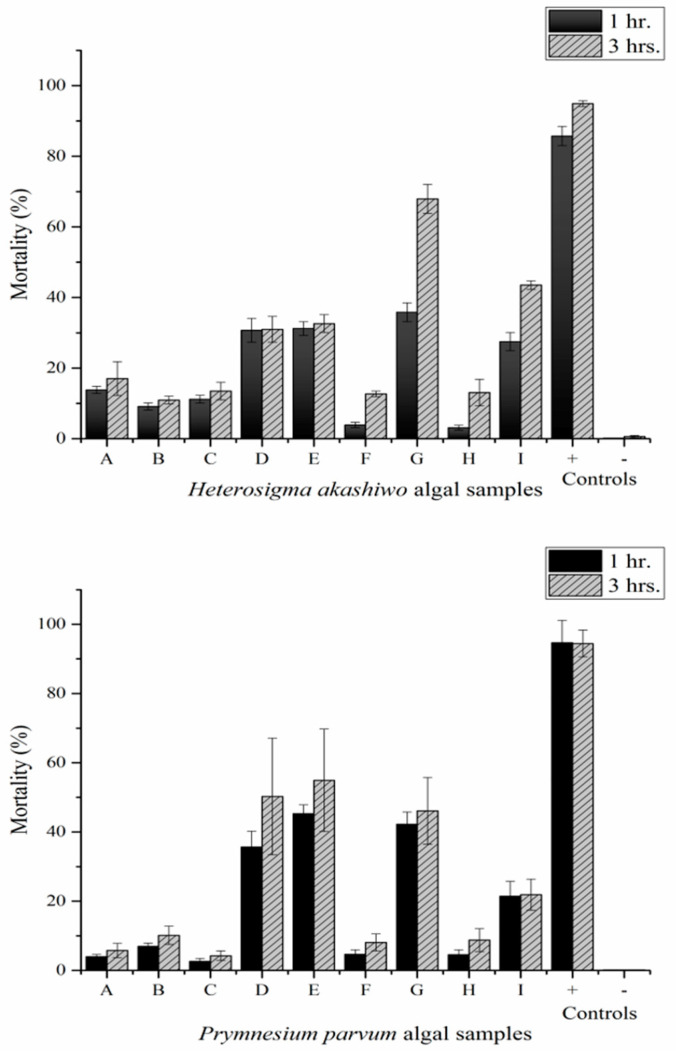
Yeast mortality (%) after exposure to different sample treatments of *H. akashiwo* and *P. parvum* (n = 3 ± SD).

**Figure 2 toxins-13-00821-f002:**
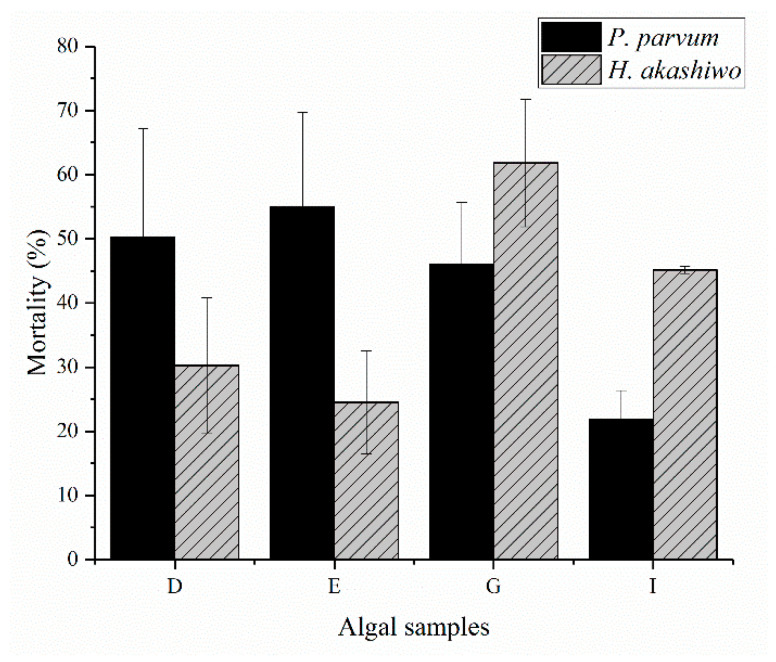
Comparison between highest yeast mortality (%) in the presence of *H. akashiwo* and *P. parvum* after 3 h incubation.

**Figure 3 toxins-13-00821-f003:**
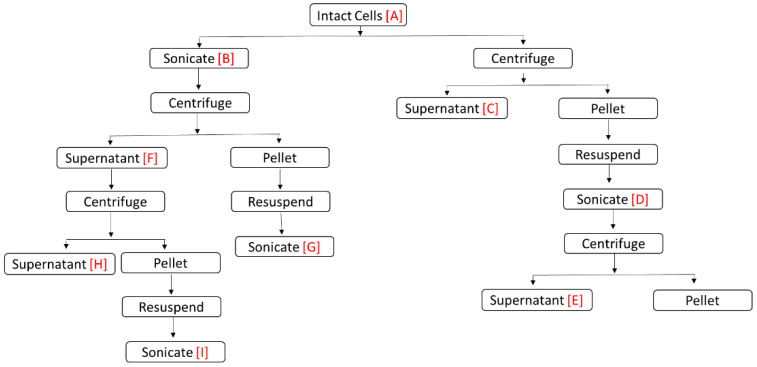
Summary of algal samples preparation.

**Table 1 toxins-13-00821-t001:** Two-way analysis of variance (ANOVA) for *H. akashiwo* and *P. parvum* toxicity measurement.

Algae	Source	Remark	Sum of Squares	Degree of Freedom	Mean Square	f-Value	*p*-Value
*Heterosigma akashiwo*	Time	^†^ Sig.	994.76	1	994.76	56.21	<0.001
	^§^ Treat.	Sig.	40,758.84	10	4075.88	230.29	<0.001
	Time × Treat.	Sig.	1358.04	10	135.80	7.67	<0.001
	Residual		778.75	44	17.7		
*Prymnesium parvum*	Time	Sig.	247.58	1	247.58	6.37	0.015
	Treat.	Sig.	51,661.38	10	5166.14	132.85	<0.001
	Time × Treat.	^‡^ N-Sig	302.93	10	30.29	0.78	0.648
	Residual		1711.04	44	38.89		

^†^ Sig = Significant; ^‡^ N-Sig = Non-significant; ^§^ Treat. = Treatment.

**Table 2 toxins-13-00821-t002:** Two-way analysis of variance (ANOVA) of the four highest effective treatments of *H. akashiwo* and *P. parvum*.

Source	Remark	Sum of Squares	Degree of Freedom	Mean Square	f-Value	*p*-Value
^†^ Algal species	Nonsignificant	1.397	1	1.397	0.0130	0.911
** Treatment	Significant	1840.645	3	613.548	5.697	0.008
Algal species × Treatment	Significant	2721.014	3	907.005	8.422	0.001
Residual		1723.203	16	107.700		

^†^ Algal species = *Heterosigma akashiwo* and *Prymnesium parvum*. ** Treatments = Sample D, E, G, I.

## Data Availability

There is no supporting data of this paper.

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
