# Peer review of "Yeast Cell as a Bio-Model for Measuring the Toxicity of Fish-Killing Flagellates"

_toxins, 2021, doi:10.3390/toxins13110821_

Round 1

Reviewer 1 Report

The authors did not show the exact mechanism of action of these toxins, they only showed that they are toxic, which is obvious. The authors suggest they have a new test. Unfortunately, this test has many problems. First of all, the authors do not specify what strain they worked on. Please remember that yeast is not a trivial model. The authors think so. What the authors forgot was the presence of a cell wall in yeast, which is variable. Therefore, in my opinion, if the authors want to publish a new method that will be used by everyone, they should check that the result is not strain dependent. Ideally, they should compare this result with animal cells. If the results are correlated, then the yeast model can be successfully used in the research. Additionally, facs are not needed for this, but a standard fluorescent microscope is enough. Additional, the problem may be the rich substrate used in your research, but also the temperature. The optimum for S. cerevisia is 28-30 oC. 
The description of the results is very poor, and it is not clear what is the control + and - .

Reviewer 2 Report

  The manuscript provide an interesting and cost efficient methods for harmful algal toxicity measuring assay. The manuscript is worth to publish after some questions are clarified.

Was there any validation process that demonstrating the results from S.cerevisiae are representative enough of the traditional cell line such as fish gill cells, etc?

Moreover, are the mechanisms of killing yeast cells similar toxicity of fish-killing?

Are different species of yeasts rather than S.cerevisiae can applied to the assay?

Lastly, if the salinity of the algal culturing medium within the supernatant may affect the mortality of S.cerevisiae in the assay? 

Reviewer 3 Report

The paper with the title "Yeast cell as a bio-model for measuring the toxicity of fish killing flagellates" reports a fast and efficient bioassay using with a yeast (Saccharomyces cerevisiae) for the measurement of the cytoxicity of two species euryhaline flagellates algaes (Heterosigma akashiwo and Prymnesium parvum).

In general, the manuscript is well structured, has a well defined objective and the results obtained are very interesting for the scientific community, providing satisfactory results on the ability of yeast as a model organism to measure the cytotoxicity caused by phytoplankton blooms.

However, I suggest the authors redrafting of the methodology made, especially the part related to obtaining the different types of samples from the initial culture of Heterosigma akashiwo and Prymnesium parvum (lines 221 to 249). Although it is true, the authors summarized the entire process in Figure 3. Also, I think the explanation of how the control samples (+) and (-) (lines 263-266) have been made should be placed right after line 249.

On the other hand, it seems that the authors have carried out the entire bioassay using a single starting culture, one for each type of algae. Could you give more information in this regard? If it had been carried out with a single culture, the results obtained would not be entirely conclusive, since its reproducibility would have to be verified. In this sense, it would also be very interesting to have calculated the LC50 for each sample.

The authors should have completed the statistical analysis by performing a component analysis of variance (PCA), although a redundancy analysis (RDA) would be much better. On the other hand, the conclusions should be more general without indicating the type of wastewater or times ...

Enter the preposition "in" on line 199 “Our results agree with previous research studies in this field. Our findings suggest that toxin(s) are in cellular compartments of the cells and cellular membrane damage can facilitate and enhance toxin release from fish-killing species.

Reviewer 4 Report

This manuscript proposed using the yeast as a model tool for screening the toxicity of bloom-forming algal species. The proposed idea is very interesting, and the manuscript shows some interesting findings. However, this work lacked a good design to demonstrate the reliability of using the yeast to screening the toxicity of bloom-forming algae. This is the biggest drawback of this work.  

To demonstrate the reliability of the new model, one typically need to validate the new one by comparing it with the traditional one. However, this work does not include such important assay. Although the manuscript indeed showed the interesting variation among different samples and between different time treatments. However, these results are hard to interpret.  Additionally, for validating the new model, one should also include the verified toxic and nontoxic algal strains in the work, and the reviewer did not see such work. 

Generally, the idea proposed in this work is certainly interesting, but the data here are too preliminary and not adequate for publishing. 

Round 2

Reviewer 4 Report

The reviewer has no further major concerns.

Author Response

Thank you for re-reading the MS. I have followed the directions and reviewed the paper for issues of grammar. The alterations are presented as "track changes."

Best Wishes.
